# Why Is the Grass the Best Surface to Prevent Lameness? Integrative Analysis of Functional Ranges as a Key for Dairy Cows’ Welfare

**DOI:** 10.3390/ani12040496

**Published:** 2022-02-17

**Authors:** Paul Medina-González, Karen Moreno, Marcelo Gómez

**Affiliations:** 1Departamento de Kinesiología, Facultad de Ciencias de la Salud, Universidad Católica del Maule, Talca 3480112, Chile; 2Programa de Doctorado en Ciencias Veterinarias, Universidad Austral de Chile, Valdivia 5110566, Chile; 3Laboratorio de Paleontología, Facultad de Ciencias, Instituto de Ciencias de la Tierra, Universidad Austral de Chile, Valdivia 5110566, Chile; 4Instituto de Farmacología y Morfofisiología, Facultad de Ciencias Veterinarias, Universidad Austral de Chile, Valdivia 5110566, Chile; marcelogomez@uach.cl

**Keywords:** animal science, movement ecology, biomechanics, dairy cows, lameness, movement analysis approach

## Abstract

**Simple Summary:**

Lameness is a highly prevalent clinical condition that causes movement disorders in dairy cows worldwide. With an estimated global population of one billion dairy cows, producing 522 million metric tons of milk per year, this problem affects food availability as well as the global economy. While grass is considered to be the best support surface for cattle, in many places it cannot be used, particularly when climate conditions are too harsh for grass to grow or be maintained. In this paper, we investigate whether grass is the best surface to prevent lameness. The answer to this question is fundamental to establishing better farming practices for cattle welfare. We built an integrative analysis of functional ranges to establish the minimum and maximum movement capacities that a cow has, according to the surfaces to which it is subjected in free housing systems. Using this analysis, we identified many aspects that make a grass surface the healthiest option for cattle. However, when grass is not available, this type of strategy can help to find the best characteristics for other possible surfaces. Our study applies movement analysis to one of the most critical problems in the world of livestock management and contributes towards finding the balance between animal welfare and production.

**Abstract:**

Lameness is a painful clinical condition of the bovine locomotor system that results in alterations of movement. Together with mastitis and infertility, lameness is the main welfare, health, and production problem found in intensive dairy farms worldwide. The clinical assessment of lameness results in an imprecise diagnosis and delayed intervention. Hence, the current approach to the problem is palliative rather than preventive. The five main surfaces used in free housing systems in dairy farms are two natural (grass and sand) and three artificial (rubber, asphalt, and concrete). Each surface presents a different risk potential for lameness, with grass carrying the lowest threat. The aim of the present study is to evaluate the flooring type influences on cows’ movement capabilities, using all the available information relating to kinematics, kinetics, behavior, and posture in free-housed dairy cows. Inspired by a refurbished movement ecology concept, we conducted a literature review, taking into account kinematics, kinetics, behavior, and posture parameters by reference to the main surfaces used in free housing systems for dairy cows. We built an integrative analysis of functional ranges (IAFuR), which provides a combined welfare status diagram for the optimal (i.e., within the upper and lower limit) functional ranges for movement (i.e., posture, kinematics, and kinetics), navigation (i.e., behavior), and recovery capacities (i.e., metabolic cost). Our analysis confirms grass’ outstanding clinical performance, as well as for all of the movement parameters measured. Grass boosts pedal joint homeostasis; provides reliable, safe, and costless locomotion; promotes longer resting times. Sand is the best natural alternative surface, but it presents an elevated metabolic cost. Rubber is an acceptable artificial alternative surface, but it is important to consider the mechanical and design properties. Asphalt and concrete surfaces are the most harmful because of the high traffic abrasiveness and loading impact. Furthermore, IAFuR can be used to consider other qualitative and quantitative parameters and to provide recommendations on material properties and the design of any surface, so as to move towards a more grass-like feel. We also suggest the implementation of a decision-making pathway to facilitate the interpretation of movement data in a more comprehensive way, in order to promote consistent, adaptable, timely, and adequate management decisions.

## 1. Introduction

Domestic cattle, *Bos taurus* and *Bos primigenius indicus*, are considered important assets for animal produce. Their metabolism transforms low-quality forage into highly energetic tissues and secretions, such as fat, muscle, and milk [1]. This effective system has been managed by humans since domestication in southeastern Turkey about 10,500 years ago [2]. Beef and dairy cow production constitute important economic activities worldwide [3,4].

The world census is more than 1.4 billion cattle heads, meaning there is roughly one cow for every five people on the earth [4]. Of these cattle heads, one billion are dairy cattle, which produce 522 million metric tons of milk per year [3], a highly appreciated nutritional beverage in human societies [1,3,4]. Therefore, there is a need to generate the necessary knowledge to understand the dynamics of animal production and welfare, aiming for a sustainable balance between the two, as Willham (1986) pointed out [5]. Cattle locomotion is recognized as an important indicator of overall animal welfare [6], and it directly influences milk production [7].

Movement analysis of individual dairy cows is frequently described in the relevant literature using the following parameters:(1)The posture of the limb, static joint angle analysis in the forelimb, hindlimb, or both combined [8,9,10].(2)Kinematics of locomotion, which is based on spatial and temporal gait parameters [11,12,13,14] and the variation of joint angles [8,9,10].(3)Kinetics at the hoof and load distribution on the contact surface with the soil (i.e., the sensory plate) and at the phalanges by using finite element modeling [15,16].(4)The behavior of the individual cow and its choice of activities, such as getting up or lying down [17,18], and the amount of displacement (using global positioning system tracking) [19].

Lameness is a painful clinical condition of the bovine locomotor system that results in alterations of movement, gait deviation, and abnormal postures. This bovine health problem determines behavior and animal welfare losses (i.e., discomfort) [20]. Lameness, together with mastitis and infertility, is considered one of the main welfare, health, and production problems in intensive dairies worldwide [21,22,23], and its economic impact has been estimated at USD 500 per cow [6]. Lameness is a multifactorial condition. It is the result of an interaction between housing design, farm management, nutrition, infectious agents, and genetic predisposition. Claw disorders account for around 90% of all lameness incidents [22,23]. There is a wide variability of lameness prevalence between farms, geographic zones, and housing systems [22]. Cook (2016) reported that the average prevalence of lameness is close to 25% for all production systems worldwide [24]. For example, in European countries, the prevalence of bovine lameness is 30%, and in the New York/Pennsylvania area of the USA, it is over 50%; while in South America, a prevalence of 30% in Brazil, 40% in Uruguay, 23% in Argentina, and 30% in Chile have been reported [21,22,23,24].

Currently, there are two ways to detect lameness in dairy cows, by visual scoring [20] and via automatized systems [25]. However, although both methods might provide an early diagnosis, these indicators are rarely used in large-scale farming [20], and they do not allow for the early detection of this dysfunction. Hence, the two detection methods do not provide a preventive solution.

There are three main causes of bovine lameness in free housing systems [9,23,26]:(i)Low stall hygiene, which increases the risk of sliding, leading to injuries, and it boosts the risk of infections [9,23];(ii)Metabolic acidosis, as a consequence of a carbohydrate-rich diet [26];(iii)Mechanical imbalance, such as an increase in high loads at the joints, and/or accelerated hooves wear, due to hard and abrasive surface stall flooring [9].

Among these causes, hard and abrasive surfaces are recognized in the relevant literature as the main factor inducing bovine lameness [27]. One clear signal that shows this is true is that about 80% of dairy cows housed at facilities with stalls that have fully slatted flooring and bare concrete solid floors are reported to be subject to at least one or more hoof disorders (i.e., line and sole diseases) [28]. Therefore, flooring type is a highly relevant factor to be considered in decisions, where the objective is to optimize the management and production process. This factor needs to be considered in order to find the balance between disease control and milk production at the lowest cost.

### Flooring Types

There are five main types of floors in free housing systems in dairy farms, and each one displays unique characteristics. In basic terms, a “good flooring surface” (i.e., the one with minimal lameness prevalence) must be neither too hard, nor too soft, too abrasive, or too slippery. In terms of the latter, flooring is best when it naturally regulates water retention by having an optimal soak range. All of these conditions affect pedal health as they affect the floor’s mechanical performance, including facilitating appropriate movement (i.e., no injuries due to sliding), the frictional level (i.e., the ability to cause dangerous abrasions), and/or the loading optimal range (i.e., injury from impacted joints). The prevalence of lameness using each of the five main flooring types can be summarized as follows.

(1)Grass presents with a 1 to 22.5% prevalence [29,30,31], and it also reduces the clinical signs of lameness in affected animals, which are transferred to this substrate type [32]. This surface prevents sliding, while allowing the homogeneous distribution of the load in the cow’s feet [8,14]. However, pasture systems can expose cows’ feet to several infectious agents, and muddy conditions could affect the incidence or prevalence of lameness in pasture-based herds. Ranjbar et al. (2016) report that the average daily rainfall is a risk factor associated with the prevalence of lameness (odds ratio: 1.06; 95% confidence interval: 1.02 to 1.09) [23].(2)Sand shows a 5 to 21.5% prevalence of lameness [33]. This surface usually maintains its dry condition, but it generates peak loads at the hooves. Hence, a higher walking effort is required by the animal [27,34].(3)Rubber has a 5 to 27.9% prevalence of lameness [35]. Its unfavorable characteristics under humid conditions make it a high-risk surface for injuries [10], but it does provide reasonably good mechanical load absorption [16]. Rubber flooring decreases slipping and the number of strides and it helps to alleviate pain and reduce wear on the feet.(4)Asphalt has a prevalence of lameness between 13.3 and 40.9% [10,36]. Under wet conditions, asphalt becomes more slippery than rubber [36]. Therefore, it is the most abrasive surface of all, and it is also very rigid, which intensifies loads [14,36].(5)Concrete presents with a 19.8 to 68.4% prevalence of lameness. It is highly slippery under wet conditions, and when covered with slurry, it is abrasive. It is also the most rigid of all of the flooring types [9,22]. An animal’s wellbeing is under the worst possible condition with this type of flooring [9,13].

For detailed information on the prevalence according to surface type (Appendix A).

Asphalt and concrete are the two surfaces with the highest levels of lameness prevalence reported. These are also the two most commonly used flooring type surfaces due to their low maintenance cost and durability, notwithstanding that this means higher losses of their valuable assets [28,37,38]. Note that locomotor disorders account for 40% of the unassisted deaths and euthanasia in dairy cows. The cows end up being severely health compromised in order to maintain production levels [37]. The aim of this study is to assess whether there is a way of reducing bovine losses caused by lameness and its associated diseases while keeping maintenance costs low.

In order to answer this question, we review the biomechanical characteristics of cow movements and posture, detecting the parameters that fall outside of the functional boundaries of all five of the different surfaces described herein [30,39]. Hence, it will be possible to improve management systems, including better-designed surfaces, in order to prevent losses, thereby improving animal welfare and production.

Functional boundaries for movement and posture are wide because the animal’s movements are ubiquitous, and its individual variability presents particular patterns of speed, angles, directions, and magnitude of forces for canonical activity modes at rest, foraging, dispersal, and migration [40,41]. Motion and navigation capacity are regulated by the internal state of the animal (i.e., its physiological and psychological condition), as well as by external environmental factors (i.e., topography, climate, the existence of predators, etc.). These components interact with each other, shaping an organism’s lifetime movement path [42]. We undertook this broad-scoped analysis using the movement ecology (ME) concept, which has linked the biomechanical and behavioral basis of movement to fitness [40,42]. This conceptual framework unifies descriptive and predictive models to determine the ecological (environmental) and evolutionary consequences of movement by addressing the questions: why, how, when, and where to move? [42]. This inspirational approach allows us to: (i) explore the effects of the substrate characteristics on the functional boundaries for movement and posture and (ii) suggest a conceptual model derived from the resultant movement parameters (i.e., the integrative analysis of functional ranges; IAFuR) to define the optimal properties that an environment should have (e.g., artificial surface design).

The present study performs an IAFuR for biomechanical parameters inspired by the ME concept (see Material and Methods) This approach could be applied to study the impact of various substrate characteristics (i.e., external factors) on movement possibilities (i.e., motion and navigation capacities for a movement path) as an indicator of the health, functionality, welfare, and production of free-housed dairy cows (i.e., their internal state). The main questions considered in this study are: (i) what is the movement range for the main flooring types used? and (ii) why is grass the best surface to prevent lameness? We hypothesized that the possibilities of movement are particular for each type of surface used, while grass has an optimal functional range for the cattle in free-housing systems. The aim of the present review is, by means of a bibliographic search, to qualitatively evaluate the flooring type influences on the movement capabilities using the posture, kinematics, kinetics, and behavior parameters of dairy cows in free-housing systems.

## 2. Materials and Methods

A literature review was carried out during the months of September 2017 and December 2018, taking into account kinematics, kinetics, behavior, and posture parameters according to the main surfaces used for free housed dairy cows. The Spanish and English language versions of Google Scholar, SciELO, Medline, and VetMed Resource search engines were used as selection criteria according to the following key concepts, in addition to the relationship between them using boolean operators: “vacas lecheras”, “dairy cows”, [and] “cojera”, “lameness” [and] “sistemas de estabulación libre”, “free housing systems” [and] “tipo de suelo”, “floor type” [and] “locomoción”, “locomotion” [or] “movimiento”, “movement” [or] “cinemática”, “kinematic” [or] “cinética”, “kinetic” [or] “conducta”, “behavior” [or] “postura”, “posture”. as selection inclusion criteria, we considered titles and/or abstracts that mentioned movement indicators and at least one of the surfaces of interest (i.e., grass, sand, rubber, asphalt, or concrete) in dairy cow free-housing systems. In addition, the publications selected had to include information on units of measurement of at least one of the indicators defined as movement parameters (i.e., posture, kinematics, kinetics, or behavior parameters; see details in Figure 1).

Of a total of 246 eligible articles, 58 met the initial selection criteria. However, only 13 of these articles presented quantitative data on movement analysis parameters according to flooring types. The information in the articles was summarized by reference to the countries in North America (*n* = 8), Europe (*n* = 3), Asia (*n* = 1), and Oceania (*n* = 1). Of these, three research papers studied grass surfaces (41 Holstein cows and 1 Fleckvieh cow), sand flooring was studied in four (36 Holstein, 1 Fleckvieh, and 340 cows from Wisconsin, USA, a breed was not specified), and rubber flooring was analyzed in 10 research papers (299 Holstein and 120 cows from Wisconsin, USA, a breed was not specified), nine investigations centered on concrete surfaces (610 Holstein cows and 1 Fleckvieh cow). Among the articles relating to concrete surfaces, two considered the humidity (soak range) generated by the accumulation of excrement. Asphalt surfaces were analyzed in five publications (260 Holstein cows and 1 Fleckvieh cow). Rubber flooring was subject to comparative analysis in five publications, with the coefficients of friction being considered as factors within the type of surface in one article (see Appendix A). Parameters used to analyze movement are detailed in Figure 1.

### 2.1. Integrating Parameters 

As the data reveal a variety of parameters all contributing to a description of cow welfare, we decided to analyze them in the light of the movement ecology concept with added considerations (Figure 2). In this integrative view, individuals can be characterized by: their internal state, their movement capacity (how to move?), and their navigation capacity (where to move?). All of these factors, but particularly the internal state, are modified by external factors, which are a series of biotic and abiotic environmental conditions that influence the movement of a given individual [42]. The internal state is a multidimensional vector that answers the question: why move? This considers the physiological and psychological characteristics of the organism in relation to the energy gain (e.g., its search for food), looking for security (e.g., escaping from predators), learning (e.g., following adults), and reproducing (e.g., finding a partner) [43,44,45].

These organism characteristics have been widely recognized by The World Organization for Animal Health (2008) [46]. This organization established that welfare means to keep animals healthy, comfortable, well-nourished, safe, and able to express innate behavior whilst not suffering from unpleasant states such as pain, fear, and distress. Fraser et al. (1997) [47], and later Von Keyserlingk et al. (2009) [48], highlighted this view specifically for dairy cows, referring to: (i) animal functioning (e.g., milk production), (ii) feelings (e.g., pain as an internal state), and (iii) the ability to live a reasonably natural life (e.g., movement and navigation capacities by reference to external factors). Therefore, all of these factors must be considered as a whole and as a single problem to be addressed in the practice of farming.

Within the framework of ME, the resting time (pause) length is considered by means of stops according to aspects of space and time, which are used to evaluate the fitness of wild animals [42]. This break time is highly important for the recovery capacity, and it raises a new question: when to move? Our approach to this question incorporates two elements that are associated with the link between recovery and internal state: (i) analysis of the energy expenditure of locomotion and (ii) its application to a pre-established context for the movement pathway of domestic animals (i.e., dairy cows in free-housing systems).

Movement, navigation, and recovery capacities are defined by the posture, behavior, kinetic, kinematic, and energetic parameters. With the results we obtained, we established functional ranges with a view to assessing wellness and health in the context of dairy cows. The scheme is seen in Figure 2.

### 2.2. Finding the Functional Ranges for the Movement Parameters in Dairy Cows

The functional ranges (either optimal or non-optimal) are a qualitative representation of the relationship between movement, navigation, and recovery capacities in which we superimpose the minimum and maximum values for the movement pathway during the life cycle, highlighting the functional boundaries achieved in the best-case situation. If the functional range is completely within these limits (i.e., within the functional boundaries), we call it the optimal functional range (i.e., the possibilities of movement on the hypothetical best-documented surface; see Graphical Abstract). A range partially outside of the boundaries is a suboptimal functional range. When the range of capacities is completely outside the functional boundaries, it is a non-optimal functional range. Although it is known that the impact of being outside the optimal range is verified in the higher prevalence of lameness (Appendix A), the purpose of the IAFuR is to determine the movement profile in each case, as well as to identify the main consequences of using alternative surfaces to grass.

Figure 3 represents a theoretical outline of animal welfare, understood as movement, navigation, and recovery adaptive responses, according to usual, fragmented, intervened, and disturbed environmental conditions, established as habitats for the study of animal dispersal [49]. It should be noted that this proposal is in an early stage of development, so for this paper, specific lines of evidence were used (i.e., movement parameters according to surface) to weigh and integrate results (Appendix A), as well as to argue the beneficial properties of alternative surfaces to grass, if required (see Section 3.2). This approach does not prevent the possibility of considering, in the future, quantitative data proposals, including additional factors such as animal size, breed, climate, and management conditions, among other documented factors [13,21,22,23].

For movement capacity, we established the natural surfaces (grass and sand) as a reference and the synthetic surfaces (rubber, asphalt, and concrete) as comparison values to answer the question of how the results were obtained for the comparison values vs. the results reported for the reference. We also considered the effect of friction and humidity as complementary factors of the synthetic surfaces. 

The functional ranges for the navigation and recovery capacities were in accordance with our qualitative interpretation of research on cattle behavior and metabolic cost (energy requirements) [50,51,52,53,54]. In dairy cows, the navigation capacity is directed by the farmer’s housing decisions. However, since these are free-housing systems, cattle preferences [50] and movement opportunities, according to surface types [51], are analyzed. These indicators indirectly affect the behavior parameter with respect to grazing, hydration, and socialization patterns. The recovery capacity is determined by an energetic cost analysis (a move–stop approach) for walking and grazing on different types of flooring [52,53] and the relationship between animal movement and the static–dynamic energy landscapes [54].

## 3. Results and Discussion

### 3.1. The Optimal Functional Ranges: Which Are the Movement Boundaries for Each Flooring Type?

The aim of the present study is to evaluate the influences of different flooring types on the movement ecology capacities using biomechanical parameters in dairy cows (Figure 2). We developed an integrative analysis of functional ranges (Figure 3) for (i) posture, kinematics, and kinetics as indicators of movement capacity; (ii) behavior as an indicator of navigation capacity; (iii) metabolic cost as an indicator of recovery capacity (Table 1).

#### 3.1.1. Why Is the Grass the Best Surface to Prevent Bovine Lameness? 

Grass is recommended for the functional recovery of healthy locomotion in dairy cows and is established as a model reference or “gold standard” [8,14,32]. It presents optimal functional ranges for movement, navigation, and recovery capacities (Table 1 and Figure 3B).

Movement capacity: Two studies evaluated movement capacity using posture, kinetic, and kinematic parameters in grass [8,14]. Herlin and Drevemo (1997) determined the angle posture of the fore and hind limb while the animals were standing. They found a trend of narrow ranges in the proximal shoulder and hip joints (10°–30°), middle ranges in the elbow, stifle, carpus, and tarsus (40°–50°), and wide ranges in the distal fetlock (80°–100°) [8]. An optimal surface allows for movement in the joint surfaces during the resting posture of dairy cows, which provides lubrication and mechanical support for the joints [55]. Alsaaod et al. (2017) report a higher speed (1.2 m/s) and “confidence” of movement when considering the acceleration at the beginning (7.8 g) and at the end (2.4 g) of the support phase on grass compared to rubber surfaces and asphalt [14].Navigation capacity: Smid et al. (2018) [50] found that cows spent more time outdoors on the grass (90 ± 6%, *n* = 12) compared with sand surfaces (44 ± 6%, *n* =12). In addition, the grass was considered, together with sand, to be the most comfortable surface for a lying, resting posture, and it provides the comparative advantage of free grazing, which is associated with optimal wellbeing [50].Recovery capacity: The grass surface has a lower locomotion cost than artificial surfaces because it causes lower stress levels (253 ± 229 N/cm^2^) compared to hard surfaces (719 ± 631 N/cm^2^, *n* = 1) [16], and it allows for greater vertical acceleration of the hoof in the move towards a standing phase (7.8 ± 0.5 g vs. 6.8 ± 0.5 g, *n* = 24) [14]. This implies that, during the support to swing transition on the grass (the sequencing of the limbs’ forces during a redirection phase), the acceleration of the push-off is more efficient and timely (before the support of the rear limb), minimizing the collisional energy losses, which conserves kinetic energy by altering the direction of the center of mass velocity vector to match a more parallel push-up of the limb [56]. This mechanism conserves the energy of the center of mass, reducing the amount of work that an animal’s muscles must perform [57].

In summary, grass is the optimal surface for free-housing systems in dairy cows because: (a) it mechanically provides a wide joint range, which boosts joint homeostasis, (b) it allows reliable and safe movement patterns while lowering metabolic costs, and (c) it promotes a longer resting time that is associated with adequate annual milk production and a lower prevalence of lameness with a range of 1% to 22.5% [29,31,36,58,59]. An interesting question that arises from these results is, do these indicators remain in poorly maintained pasture compared to well-managed indoor systems? Although we do not have direct evidence about the movement parameters according to different types and quality of grass, if we hypothetically analyze the mechanical properties of the surface, poorly maintained pasture would lose the optimal balance of the hoof horn wear and growth, in addition to the natural claw load [14]. Therefore, the grass surface property, in this case, could resemble asphalt (i.e., higher abrasion) affecting the claw conformation [12], as well as concrete (i.e., lower load distribution), causing slipping and “stiff” gait [9,10].

Despite this, overall, it presents the best indicators of wellbeing (Figure 4A). However, grass is not always available as a surface for free-housing systems, either for climatic or economic reasons, so it becomes important to have alternative surfaces that could obtain as much as possible the best standards of animal welfare, while also improving production benefits and costs.

#### 3.1.2. Sand Is the Best Alternative Natural Surface, but it Presents High Metabolic Cost and Management Challenges

Movement capacity: It results in being very similar to grass, because it allows a natural locomotor behavior, efficient strides, and no significant slipping risk [60,61].Navigation capacity: On a sand surface, only the stand-up/lie-down locomotive behavior of dairy cows is considered as an indicator of navigation capacity, because it answers the questions: when and where to move? [42]. Contrary to the normal condition found in the ME concept [42], the animals certainly are not really free to move wherever they want; indeed, they are kept in a closed environment. Therefore, most spatial distribution is led and controlled by the farmer, but even within this restriction, the animal still can decide to stand up or lie down. Sand surface presents, along with grass, the best conditions, having a longer time in a lying position posture (12.4 ± 7 h/day vs. rubber 10.7 ± 5 h/day, *n* = 208) and less time in a standing posture (11.5 ± 6 h/day vs. rubber 13.2 ± 6 h/day, *n* = 208) [61], as well as equivalent lying/standing time parameters to the grass “gold standard”. A greater time of lying behavior optimizes milk production through increases in the blood diffusion at the udder (around 5 L/min) compared with a standing animal (around 3 L/min), plasma concentration of the growth hormone, rumination frequency (reducing ruminal acidosis), and avoidance of chronic stress in the animal, according to the negative changes in the response of the hypothalamic–pituitary–adrenal axis [62,63,64]. The longest standing times are associated with the appearance of pedal pathologies, consequent of lameness [65,66].Recovery capacity: Sand is a substrate, which reduces the hoof output kinetics; hence, the animal requires a greater effort to break the inertia when walking. Dijkman and Lawrence (1997) [52] reported that the energy expenditure for the locomotive work of cattle and buffalo can be two times higher on incompetent substrates (i.e., mud) vs. concrete (3.34 J/m/kg vs. 1.69 J/m/kg, *n* = 6). This higher locomotion effort limits, in the long term, the capacity to travel long distances, because that requires greater muscle activation [44,54,57,67]. That situation triggers suboptimal ranges in kinematics and metabolic indicators (Figure 4C).

Sand appears to be the best natural alternative housing surface next to grass, as it presents excellent results in navigation and acceptable ranges in movement capacity (Figure 4C). Accordingly, the prevalence of lameness, from 5% to 21.5%, is nearly as low as grass [33,36,61,68,69,70]. Nevertheless, it presents the inconvenience of causing a higher metabolic cost for movement.

With respect to management, however, implementation poses strategical challenges. The frequent need for sand renewal implies finding solutions for good access to natural sources of sand, and it requires special manure handling, which is time/money consuming [60,61].

Thus, it is relevant to continue to search for artificial alternatives that could mimic the benefits of the natural surfaces on cattle wellbeing. This analysis also provides crucial information for the future engineering of artificially improved soil surfaces.

#### 3.1.3. Rubber Is an Acceptable Artificial Surface, but the Friction Property Is a Notable Weakness

Ten investigations evaluated movement parameters on rubber surfaces (Table 1), including posture [10], kinematics [10,12,14,34,71,72], and kinetics [14,16,73].

Movement capacity: The increased friction did not affect limb posture during the support phase of the walking cycle [10]. Though, the speed appeared lower than that recorded on sand (1.01 ± 0.02 m/s vs. 1.12 ± 0.02 m/s, *n* = 36) [34] and also on grass (1.1 ± 0.01 m/s vs. 1.2 ± 0.01 m/s, *n* = 24) [14]. Although these differences were statistically different, in all cases, the speed exceeded the documented biological threshold (>0.97 m/s) [34]; in addition, the stride length presented an optimal functional range on all surfaces (Appendix A). The optimal rubber friction range was 0.4 < μ < 0.5, but over the 0.5 value, speed decreased even more (0.81 ± 0.05 m/s vs. 0.85 ± 0.05 m/s, *n* = 5) [10]. The overall stress accumulation in the hoof was similar when walking on rubber to the one found on concrete [73]. In addition, the rubber registered lower acceleration magnitudes when unloading body weight on limbs than the ones registered in grass (6.2 ± 0.5 g vs. 7.8 ± 0.5 g, *n* = 24) [14]. This reflects how the cow´s gait behavior was being carried out under “Low Confidence”, as it is referred to in the literature. Surely, this unnatural feeling is produced by a proprioceptive reaction response to substrate competence (i.e., different mechanical properties) [67]. However, the step efficiency remained equivalent to the one observed on natural flooring types because the thoracic/pelvic foot overlap was found to be similar to sand (0.2 ± 1.5 cm vs. 1.7 ± 1.5 cm, *n* = 36) [34], and the stride length was greater than on concrete (1.55 ± 0.05 m vs. 1.4 ± 0.05 m, *n* = 645) and asphalt (1.55 ± 0.05 m vs. 1.48 ± 0.07 m, *n* = 645) [71].Navigation capacity: It was assessed through behavioral studies [24,27,62]. The lying time, as well as the frequency of postural transitions, was found to be higher than on concrete surfaces (12.3 ± 0.3 h/day vs. 10.4 ± 0.4 h/day, n = 16) [62]. However, it presented a suboptimal functional range for the slightly shorter time for lying (11.7 ± 0.2 h/day vs. 12.1 ± 0.2 h/day, *n* = 120), milking (2.6 ± 1 h/day vs. 3.2 ± 1 h/day, *n* = 120), feeding (4.1 ± 1 h/day vs. 4.7 ± 1 h/day, *n* = 120), and drinking (2.3 ± 0.5 h/day vs. 2.4 ± 0.5 h/day, *n* = 120), while showing a slightly higher standing time compared to sand (12.4 ± 4 h/day vs. 12 ± 3 h/day, *n* = 120) [24,27], Table 1 and Appendix A). From the biological point of view, these results guarantee a longer rumination time compared to hard and abrasive surfaces (i.e., asphalt and concrete) [53]. In addition, a functional range for welfare and production indicators was shared (Appendix A and Figure 4).Recovery capacity: The rubber friction property played a relevant role on locomotion. Faced with a low coefficient of friction, μ < 0.4, the gait becomes unstable, so the cow must increase cadence with a shorter stride length to maintain a certain speed [10] (Appendix A). The increase in friction generated an increased swing phase of the gait but decreased the accumulation of elastic potential energy to develop the next step [44,57]. Therefore, friction outside the optimal range generated a higher metabolic walking cost, and it was less safe.

The bovine lameness prevalence range on rubber surfaces is between 5% and 27.9% [22,35,68,69,70,74,75]. Most studies report values around 20% (Figure 4A, Appendix A) as well as a lower annual milk production than that of sand (11027 ± 240 kg/cow/year, *n* = 119 vs. 11785 ± 240 kg/cow/year, *n* = 89) [24], with no significant difference to the one reported for hard and abrasive surfaces such as asphalt (7535 ± 745 kg/cow/year, *n* = 193 vs. 7286 ± 1778 kg/cow/year, *n* = 239) and concrete (7535 ± 745 kg/cow/year, *n* = 193 vs. 7889 ± 1179 kg/cow/year, *n* = 213) [71].

This artificial surface makes bovines less confident [76]. Despite this, rubber remains an interesting alternative surface to the natural ones, if the following considerations are observed: (i) providing optimal mechanical properties of friction, stiffness (Young’s modulus), and deformation, and (ii) landscape design with elements that help to familiarize the animal (for more details see Section 3.2.1. Surface Material Properties).

**Table 1 animals-12-00496-t001:** Summary of the reviewed literature, the methodology, and the main findings in the light of the integrative analysis of functional parameters.

Reference	Methodology				Main Findings
	N	Between- Group Besign	Floor Types (Time)	Movement Parameters (Measurement Units)	
[8]	10	Independent	Grass (4) vs. concrete (6).(30 weeks)	ROM at the shoulder, elbow, carpus, MCP, hip, knee, tarsus, and MTP (degrees).	OFR for shoulder, carpus, hip, stifle, and tarsus posture on concrete. Sub-OFR for elbow, MCP, and MTP posture on concrete.
[9]	6	Repeated	Soak increment on concrete: dry vs. wetted vs. shallow slurry vs. deep slurry.(2 weeks)	Speed (m/s), step length (m), and cadence (step/min). ROM at the elbow, carpus, MCP, knee, tarsus, and MTP (degrees).	Sub-OFR for all posture parameters, speed, step length, and cadence on soak concrete. Non-OFR for step length and cadence on shallow slurry concrete. Non-OFR for speed on shallow slurry concrete. Non-OFR for speed, step length, and cadence on deep slurry concrete.
[10]	5	Repeated	Friction (μ) increment on rubber: μ =0.33 vs. μ =0.74.(2 weeks)	Speed (m/s), step length (m), and cadence (step/min). ROM at the elbow, carpus, MCP, knee, tarsus, and MTP (degrees).	Sub-OFR for elbow, carpus, MCP, stifle, and MTP posture, speed, step length, and cadence on high friction rubber. Non-OFR for step length and cadence on high friction rubber.
[13]	30	Repeated	Concrete (dry) vs. concrete (soak).(20 weeks)	Stride length (m), asymmetry for step width (m), asymmetry for step length (m), and overlap (m).	Sub-OFR for overlap on soak concrete. Non-OFR for stride length, asymmetry for step width, and asymmetry for step length on soak concrete.
[14]	24	Repeated	Grass vs. rubber vs. asphalt.(48 weeks)	Speed (m/s), stride length (m), foot load, and toe-off confidence (g).	OFR for stride length on rubber and asphalt. Sub-OFR for toe-off confidence on rubber. Non-OFR for speed and foot load confidence on rubber and asphalt.
[16]	1	Repeated	Soft (rubber) vs. hard (concrete).	Stress total (N/cm^2^). (Dissected limb model)	Non-OFR for stress total on concrete.
[27]	120	Independent	Sand (60) vs. rubber (60).(52 weeks)	Time of lying, standing, milking, feeding, and drinking (hours/day).	Sub-OFR for lying, standing, milking, feeding, and drinking time on rubber.
[34]	36	Repeated	Sand vs. rubber vs. concrete(instant response, 10m walkways)	Speed (m/s), step length (m), stride length (m), and overlap (mm).	Sub-OFR for step length and stride length on rubber and concrete. Non-OFR for speed and overlap on rubber and concrete.
[61]	208	Independent	Sand (89) vs. rubber (119).(36 weeks)	Time of lying, standing, milking, feeding, and drinking (hours/day).	Sub-OFR for lying, standing, milking, feeding, and drinking time on rubber.
[71]	645	Independent	Rubber (193) vs. asphalt (239) vs. concrete (213).(24 weeks)	Stride length (m).	Non-OFR for stride length on asphalt and concrete.
[72]	40	Repeated	Rubber vs. asphalt vs. concrete.(20 weeks)	Speed (m/s), stride length (m), and asymmetry for step length (m).	Sub-OFR for speed on concrete and stride length on asphalt and concrete. Non-OFR for speed and step length asymmetry on asphalt, and step length asymmetry on concrete.
[73]	45	Independent	Rubber (16) vs. asphalt (16) vs. concrete (13).(27 weeks)	Pressure at the claw and foot (N/cm^2^).	Sub-OFR for pressure at the claw on concrete. Non-OFR for pressure at the claw on asphalt. Non-OFR for pressure at the foot on asphalt and concrete.
[77]	16	Repeated	Rubber vs. concrete.(3 weeks)	Time of lying and standing (hours/day). Frequency of lying and standing (times/day).	Sub-OFR for standing time on concrete. Non-OFR for lying time, lying frequency, and standing frequency on concrete.

OFR: optimal functional range as contrasted with grass “the gold standard”; sub-OFR: sub-optimal functional range; non-OFR: non-optimal functional range. MTC: metacarpophalangeal joint (forelimb fetlock angle); MTC: metatarsophalangeal joint (hindlimb fetlock angle); μ = friction coefficient. Between-group design: (i) repeated: the same individuals were evaluated on each surface; independent: different groups of individuals were evaluated between each surface. For the definition of movement parameters, see Figure 1. For graphic details, see Appendix A.

#### 3.1.4. Asphalt Is a Harmful Surface: Highly Abrasive for the Hooves

Four studies were found that analyzed the movement capacity using indicators of kinetics [73] and kinematics [14,71,72].

Movement capacity: The asphalt surface presented the lowest stress values at the hoof and foot compared to rubber (40 ± 2 N/cm^2^ vs. 57 ± 2 N/cm^2^, *n* = 16) and concrete (40 ± 2 N/cm^2^ vs. 66 ± 2 N/cm^2^, *n* = 13) [73]; in addition, a lower “confidence” was reported vs. grass, measured by the acceleration of the foot during the support phase (6.8 ± 0.5 g vs. 7.8 ± 0.5 g, *n* = 24) [14]. This situation can be explained by proprioceptive mechanisms, which trigger the animal to walk with greater caution and insecurity, decreasing speed (1.3 ± 0.5 m/s vs. 1.4 ± 0.5 m/s, *n* = 40) [14,72] and stride length (1.48 ± 0.05 m vs. 1.55 ± 0.05 m, *n* = 645) [71], in addition to increasing the variability in stride length (4.3 ± 0.5 cm vs. 4 ± 0.5 cm, *n* = 40) [72]. The data reported for the movement capacity, added to the high abrasive component, vs. rubber and concrete surfaces, modifying hooves conformation (reduced claw sole concavity), greater rates of claw wear (5.3 ± 0.3 mm/mo vs. rubber 1.4 ± 0.2 mm/mo vs. concrete 1.6 ± 0.3 mm/mo, *n* = 23), and lower claw net growth (−0.2 ± 0.4 mm/mo vs. rubber 2.5 ± 0.2 mm/mo vs. concrete 2.5 ± 0.4 mm/mo, *n* = 23) [12], supporting the inference that it is an uncomfortable surface for the cow.Navigation capacity: We can only hypothesize that cows may have a reduced lying time, and an increased standing time, since they were not directly recorded in the investigations found [62,63,64]. However, compared to rubber, a higher alteration of rumination times and milk production have been reported, which point to a suboptimal lying/standing time (7286 ± 1778 kg/cow/year vs. 7535 ± 745 kg/cow/year, *n* = 645) [71].Recovery capacity: When walking with greater caution, the metabolic cost is higher, since the optimal speed threshold would not be reached [44], negatively affecting the recovery capacity (Figure 4E). It would be interesting for future research to quantify the impact of this surface on the specific indicators of behavior and movement cost, in order to evaluate the magnitude of this impact.

The alteration on movement parameters that this surface generates, given its high abrasion and the discomfort it causes to the animal, makes the use of this surface in free housing of dairy cows not recommended. This situation is confirmed with a high prevalence of lameness, which fluctuates between 13.3% and 40.9% [33,35,36], Figure 4A.

#### 3.1.5. Concrete Is a High Impact and the Most Damaging Surface

Nine investigations analyzed the effect of a concrete surface on movement, navigation, and recovery capacities [8,9,12,13,16,71,72,73,77].

Movement capacity: Concrete has been studied according to differences in soak range. Walking speed (0.65 ± 0.05 m/s vs. 0.81 ± 0.05 m/s, *n* = 6), stride length (1.6 ± 0.01 m vs. 1.7 ± 0.01 m, *n* = 30), and cadence (0.39 ± 0.05 steps/min vs. 0.58 ± 0.05 steps/min, *n* = 6) decreased as the floor soak increased [9,13]. These indicators favored the appearance of foot lesions, hemorrhages, dermatitis, and sole erosions, which present lameness as a functional consequence [74,75]. There is a high metabolic cost of locomotion, which makes them prone to pedal lesions given the repetitive mechanical impact over time. The posture was analyzed in two European studies developed in Holstein cows (Table 1). Suboptimal functional ranges compared to grass were observed on joint stiffness at the hip (198 ± 6° vs. 201 ± 4°, *n* = 17), metacarpo–phalangeal joint (60 ± 60° vs. 166 ± 60°, *n* = 17), and metatarso–phalangeal joint (171 ± 55 vs. 162 ± 48°, *n* = 6; 186 ± 24° vs. 183 ± 25°, *n* = 17) [8].

A slight negative impact was observed when a higher level of humidity and depth given by the amount of slurry was added: metacarpo–phalangeal joint (176 ± 20° vs. 174 ± 20°, *n* = 6) and metatarso–phalangeal joint (171 ± 55° vs. 162 ± 48°, *n* = 6) [9], Table 1. Indeed, numerical values did not show significant differences, but clinically, it affected the aplomb of the animal. Vermunt and Greenough (1994) [66] established that, during weight bearing, a minimal decrease in the angle of the knee and the tarsus generated joint stress increase. A straight hock posture is an adaptive mechanism to avoid slipping on soak surfaces [9].

Among the kinetic variables, the evaluation of the contact effect of the hoof on different surfaces stands out, reporting that hard surfaces such as asphalt and concrete have axial stress up to three times higher than flooring considered soft, such as grass and sand [16]. This situation is mainly due to the smaller contact surface, increasing the pressure at the hooves (65 ± 4 N/cm^2^ vs. 53 ± 4 N/cm^2^, *n* = 178) [73].

The question remains, which hard artificial surface is the best? Telezhenko et al. (2008) [73] reported that when the asphalt presents a μ near the ideal (0.4–0.5), it represents an advantage over concrete because the vertical ground reaction force distribution at the hoof zones is more homogeneous on asphalt (lateral claw 56 ± 3.7%; medial claw 44 ± 3.7%, *n* = 16) vs. concrete (lateral claw 65 ± 4.3%; medial claw 35 ± 4.3%, *n* = 13). Haufe et al. (2009) [71] reported that, in asphalt, the stride length is significantly higher (1.48 ± 0.07 m vs. 1.4 ± 0.05 m, *n* = 645), making asphalt preferable over concrete since it boost a more secure and efficient locomotion.

Navigation capacity: The concrete surface registered the shortest lying down time (10.4 ± 0.4 h/day vs. 12.3 ± 0.3 h/day, *n* = 16) and longer standing up times (12.4 ± 4 h/day vs. 12 ± 3 h/day, *n* = 16) when compared to rubber surfaces [77]. These two behaviors negatively compromise the welfare and production indicators, due to the decrease in rumination time, increased lactic acid metabolism, stress, and high mechanical pressure on the hooves, which is sustained over time and is a potential risk for hoof integrity [53].Recovery capacity: Movement indicators suggest poor performance on a concrete surface, a situation that worsens with slippery floors (increased soak). A non-optimal functional range in the overlap feet parameter (37 ± 1 mm vs. 171 ± 1 mm, *n* = 36) [34], stride length (1.48 ± 0.07 m vs. 1.4 ± 0.05 m, *n* = 645) [71], and cadence (0.39 ± 0.05 steps/min vs. 0.58 ± 0.05 steps/min, *n* = 6) [9], and a suboptimal step length variability (3.4 ± 0.5 mm vs. 4.1 ± 0.4 mm, *n* = 40) [72]. These parameters cause a negative impact on joint health [51]. It is difficult to conserve the body and limbs’ mechanical energy, as well as their energy exchange between potential, kinetic, and elastic energies [57].

Concrete is considered one of the most commonly used types of surfaces in livestock management due to its long-term durability and ease of cleaning [11,12]. However, on this surface, the highest prevalence of lameness is reported (ranging from 19.8 to 68.4%) [22,29,35,58,59,69,74]. Several claw disorders are observed more frequently under this type of flooring. Overall, this surface shows the worst indicators of movement, navigation, and recovery capacities; it generates insecurity, high metabolic cost, and hoof stress, Figure 4A.

### 3.2. The Integrative Analysis of Functional Ranges Provides Directions for an Artificial Surface Engineering 

In places where there is high weather variability (either too cold/hot, or too dry/wet), it is impossible to implement a grass surface [78]. Therefore, it is useful to think about the conditions set at the very base of the refurbished Movement Paradigm, the so called “External Factors” (i.e., substrate quality), in order to achieve a protocol established on movement parameters for adaptable decision making. Optimal functional ranges of movement, navigation, and recovery capacities can be approached by specific modification of the substrate; this allows an improvement of the animal “Internal State” (Figure 2, refurbished Movement Ecology Paradigm). The present study, based on the current available information (Results and Appendix A), indicates the best mechanical, design, and handling property choices to take into account for artificial surface design. 

#### 3.2.1. Surface Material Properties

The recommended elasticity for an artificial surface allows a deformation between 3.3 mm for 250 kg [34] and 10 mm for 400 kg [79]. The friction coefficient should be set between 0.4 and 0.5 μ, as in rubber [10,79], allowing a safe and efficient gait with the least wear on the hoof [72,80].

Heat flux inherent material property must be thoroughly evaluated as well, because heat balance depends on the metabolism of the animal, the climatic conditions, and the time of exposure to the substrate [54]. Heat stress, at an initial stage, causes an increased grazing movement, but then there is later activity decrease due to exhaustion [81], thereby affecting recovery capacity. Ruunaniemi et al. (2005) [79] obtained data for seven types of rubber that showed a wide material performance. Some of these materials were best for cold climate (seasonal) because they lost heat slowly, while others were best for hot weather because the heat flux was as much as three-fold faster. Hence, it is important to choose well which material and for how long it is best to implement it.

Future research should focus on the interaction of friction, elasticity, and heat flux on the animal’s internal state and its various movement parameters. Tuning these properties would also reduce erosion due to high traffic, while allowing an optimal functional range for movement (i.e., posture, kinematics, and kinetics) and navigation capacities (i.e., lying down position) [8,14,27].

#### 3.2.2. Desired Surface Design

Surface design must facilitate an adequate ability to clean and durability [82]. It has to include an effective drainage system (e.g., slope control and well-placed sand/gravel belts) to keep an appropriate soak range (graphical abstract). The construction of a thorough surface plan design will allow a safe and efficient movement of the animal in free-stalls: preventing falls, improving metabolism, reducing diseases, ameliorating behavioral stress, etc. This results in an overall improvement in cow welfare (i.e., movement pathway) leading to a better production (Figure 5) [9].

## 4. Conclusions

In the present work, we constructed an integrative analysis of functional ranges (IAFuR), based on an adapted version of the Movement Ecology Paradigm, to evaluate the movement parameters involved in the risk of lameness in cows on five different surfaces used in free-housing systems: two natural (grass and sand) and three artificial (rubber, asphalt, and concrete). Grass provides the optimal functional range, as the so called “gold standard floor”. Sand is the best natural alternative surface but presents elevated metabolic expenditure and managing cost. Rubber is an acceptable artificial alternative surface, but it is important to consider the mechanical and design properties. Asphalt and concrete are the most harmful surfaces due to the high abrasiveness and loading impact.

In light of the IAFuR, we were also able to provide recommendations to improve the material properties and design of an artificial surface, taking into account the elasticity, friction coefficient, and heat flux, as well as adequate visual and proprioceptive characteristics that facilitate the animal’s movement confidence given by a grass-like feel. All of these ideal characteristics improve cow welfare and will, as a consequence, enhance its production. 

Movement parameters, as well as any other either qualitative or quantitative measurable factor, can be analyzed interacting together in the IAFuR. Therefore, assessments on the overall animal pathway (movement through the animal’s life cycle) considering the random approach [42,83,84,85], could also be integrated to the IAFuR.

Limitations of the study are the reduced number of analyzed studies that met the biomechanical inclusion criteria and variations in the studies design and baseline characteristics of the animals involved. Further experimental comparison research is needed for adequate evaluation of the biomechanical properties of the different flooring types used in dairy farms.

In spite of this, and in a more general perspective, IAFuR can be used to tackle multifactorial research/clinical problems, including complex environmental factors in: (i) applied sciences such as veterinary medicine, rehabilitation, occupational science, and sport; (ii) ecophysiological and paleobiological interpretations; (iii) the development of innovative engineering and biomimetic materials.

## Figures and Tables

**Figure 1 animals-12-00496-f001:**
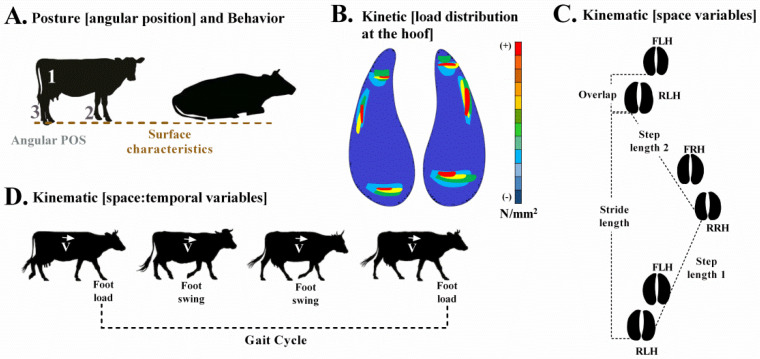
Schematic view of animal posture, behavior, kinetic, and kinematic parameters. (**A**). Posture and behavior analysis. Behavior is evaluated in two phases. First, when the animal is in a standing position, three posture parameters are recorded: (1) hip angle position relative to the back, (2) joint angle at the wrist, and (3) joint angle at the ankle. Second, measurements were taken as the time budget used for lying down in a resting position against the time standing up (hours per day). (**B**). Kinetics are estimated by the load distribution through the whole hoof. This could be measured with a force platform or modeled using a computer simulation (Finite Element Analysis), both expressed as Von Misses stress (N/mm^2^) (illustrative image modified from Hinterhofer et al., 2005) [16]. (**C**). Kinematics of the stride are characterized by its length (distance of one gait cycle, in meters), step length (distance between the hind or anterior right leg vs. the left, in meters), and anterior vs. posterior leg overlap, which, if positive, indicates that during a stride, the advance of the anterior limb in relation to the posterior limb on the same side is greater than 0 cm. FLH: front left hoof; RLH: rear left hoof; FRH: front right hoof; RRH: rear right hoof (modified from Telezhenko, 2009) [11]. (**D**). Kinematics of the gait cycle are established by the gait speed (distance per unit of time, in m/s), the cadence (steps per unit of time, in steps/s), and the time of the support and balancing phases during the gait cycle. Source: own illustration based on the articles cited.

**Figure 2 animals-12-00496-f002:**
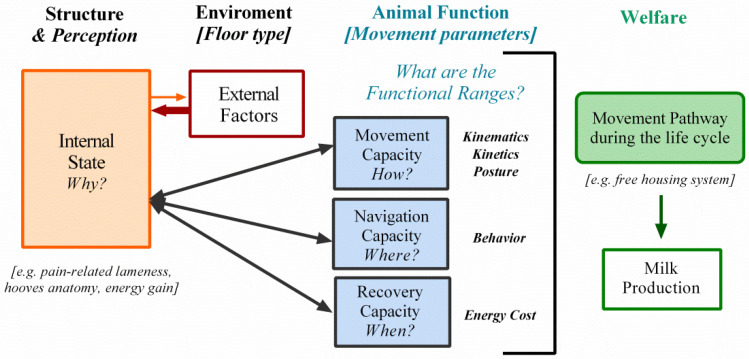
Bovine lameness was analyzed using a refurbished movement ecology concept methodology (modified from Nathan et al., 2008) [42]. This perspective allows a comprehensive view that can be assessed by functional parameters that contribute to bovine welfare, which aids production. Both the anatomical structure of the hoof and the animal’s psychological and physiological condition are defined as the internal state, which, in turn, is modified by, and introduces certain modifications to, the external factors. In particular, floor characteristics (an external factor) have an important effect on the cattle internal state (see thick arrow), exemplified here in the prevalence of pain-related lameness. Animal droppings can affect the surface frictional and soak status of the flooring, modifying the external factors to a lesser degree. Movement parameters (manifestations of the cow’s internal state) define the movement, navigation, and recovery capacities, and are indicators of function. To answer the question: why is the grass the best surface to prevent lameness? We explore the functional boundaries of each of these parameters according to the floor type. In sum total, along the animal’s life cycle, and as seen in the current study that focuses on housing systems, a movement pathway is determined, ensuring the animal’s welfare and conditioning its milk production capacity. Source: own creation inspired on the article cited.

**Figure 3 animals-12-00496-f003:**
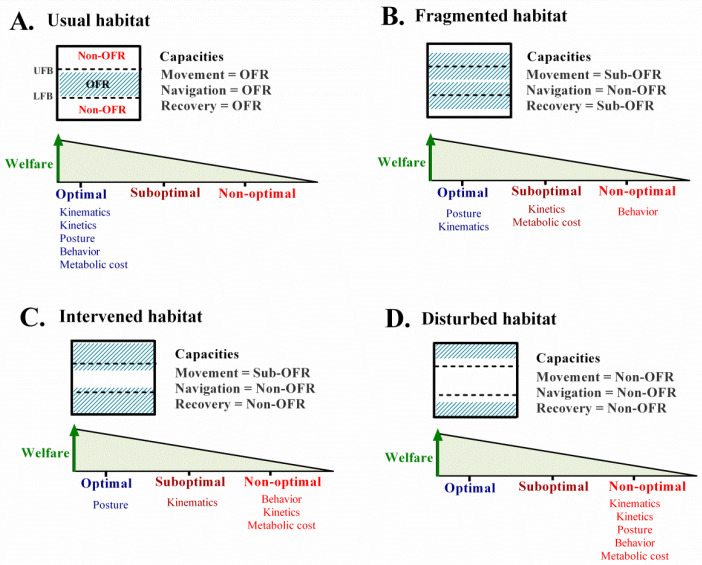
The methodology used for the integrative analysis of the functional ranges for movement, navigation, and recovery capacities according to different environmental conditions. The functional integration diagram for welfare status (above) and the quality of the movement parameters (below) are shown for each environment. (**A**). Usual habitat. Food resources, places of protection, and usual environmental conditions in which all capacities are completely within the functional boundaries, which is understood as the optimal functional range (OFR). (**B**). Fragmented habitat. There are sufficient resources and adequate protection sites; however, these are dispersed in the habitat. The navigation capacity is non-OFR. The movement capacity is in the OFR with respect to posture and kinematics and in the non-optimal functional range (non-OFR) with respect to kinetics. This is called sub-OFR. The recovery capacity presents as within the sub-OFR. (**C**). Intervened habitat. There are limited resources, places of protection, and adequate environmental conditions. The navigation and recovery capacities present are in the non-OFR. The movement capacity is in the sub-OFR; in the Non-OFR for kinematics, kinetics, and metabolic cost; in the OFR only for posture. (**D**). Disturbed habitat. There are limited resources, scarce places of protection, and unpredictable environmental conditions. The movement, navigation, and recovery capacities are present in the non-OFR. The dotted lines represent the upper (UFB) and lower functional boundaries (LFB).

**Figure 4 animals-12-00496-f004:**
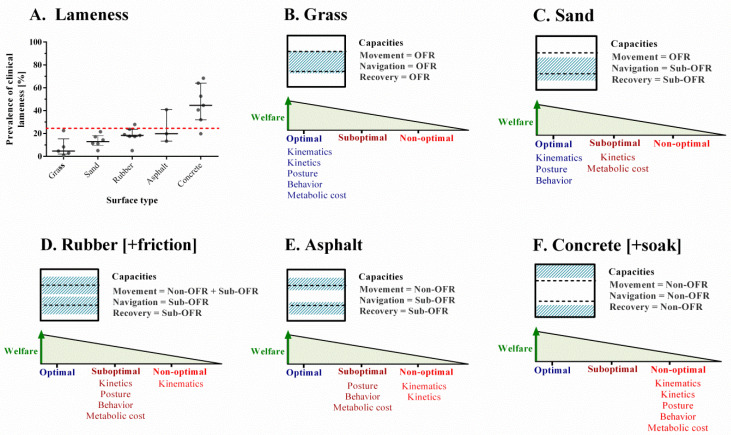
Integrative analysis of functional ranges for postural, behavioral, and locomotion indicators in dairy cows according to different types of surfaces evaluated. (**A**). A graphical representation of the lameness prevalence for each type of surface, as found in the literature, using a 95% confidence interval plot; average seen as a bold line. Points represent the result recorded for each study (see Appendix A). The red dotted line represents the average prevalence of reported lameness [24,36]. Welfare status diagrams for: (**B**). grass, (**C**). sand, (**D**). rubber plus friction increment, (**E**). asphalt, (**F**). concrete plus soak increment. The optimal functional range is represented by the dashed lines and the functional range of each surface by the hatched area. The profile for the movement indicators for each type of soil is shown below each scheme. OFR: optimal functional range; sub-OFR: suboptimal functional range; non-OFR: non-optimal functional range.

**Figure 5 animals-12-00496-f005:**
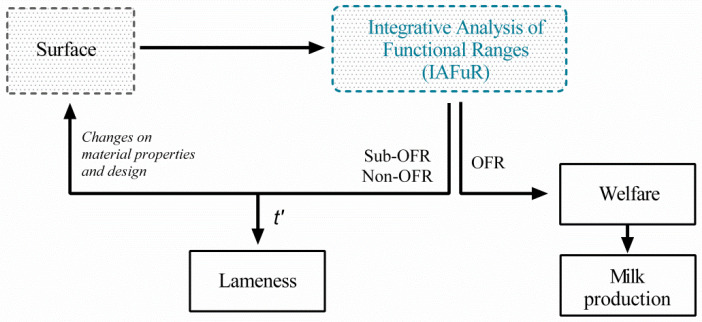
Decision-making processes in bovine lameness by an adaptable strategy based on an integrative analysis of functional ranges. The conventional surfaces’ (i.e., grass, sand, rubber, asphalt, and concrete) impact on the movement parameters (i.e., posture, kinematics, kinetics, and behavior) is evaluated by the IAFuR. The information of movement capacities drives two possibilities of decision making: (i) for optimal functional ranges of all movement capacities, a state of wellbeing with high production is favored and (ii) for suboptimal and non-optimal functional ranges, it is necessary to develop a new alternative surface by evaluating specific material and design properties for the animal’s requirements. This external factor changes (i.e., surface’s engineering) will modify the internal state (i.e., lameness risk over time), capacities, and pathway of the animal’s movement, which will be evaluated by a new IAFuR decision-making process. OFR: optimal functional range; sub-OFR: suboptimal functional range; non-OFR: non-optimal functional range; t’: sub-OFR and non-OFR over time.

## Data Availability

All the data generated or analyzed during this study are included in this published article (and its Appendix A).

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
