# Peer review of "Why Is the Grass the Best Surface to Prevent Lameness? Integrative Analysis of Functional Ranges as a Key for Dairy Cows’ Welfare"

_animals, 2022, doi:10.3390/ani12040496_

Round 1

Reviewer 1 Report

The manuscripts by Medina-González et al. describe the relationship between four different flooring types and the movement capacities in dairy cows. The authors used an Analysis of Functional Ranges for posture, kinematics, and kinetics (indicators of movement, behavior, and metabolic capacities). One of the main contributions of the study is the improvement of the welfare and productivity of dairy cows. Furthermore, the knowledge obtained through the review is crucial to cattle producers in general, as lameness is one of the main welfare and health issues faced by the beef industry as well. A strength of the manuscript is the fact of using a large range of indicators to assess the effects of both natural and artificial flooring types on cow's welfare. Overall, the manuscript is very well written. The subject is important and worth reporting, however, in the current form, I find the manuscript a bit superficial and report-like. I encourage the authors to discuss deeper and be critical about the findings of the previous studies. In addition, the authors should describe the potential weaknesses in the study due to the lack of previous studies/references approaching this topic. Below are some suggestions for improvement.

1- L139 - please add a reference after: " It allows for a virtually..."

2- L142 - please add a reference to the following sentence: "Surface maintains its dry..."

3- L143 - please add a reference to the following sentence:" A higher walking effort is overcome by the animal."

4- L144 - please add a reference to the following sentence: "Its unfavorable compliance under..."

5- L147 - please add a reference to the following sentence: "Under wet conditions the asphalt..."

6- L152 - please add a reference to the following sentence: "Animal well-being is under the worst conditions."

7- L162 - check for double space in the following sentence and throughout the entire manuscript: "____With this information..."

8- L181 - " ii) why the grass is the best surface to..." Is this the hypothesis of the study? If yes, please make sure this is clear to the reader. Also, I would like to suggest to the authors to explore more in the introduction section why you believe the grass is the best one by using previous studies as references. 

9- How concerned are the authors about having such low numbers (n = 13) of studies/references that followed their criteria of selection?

10- L444-447 - the authors should add a discussion about the fact that although the speeds described in the previous studies/references were statistically different (if that is the case), biologically they seem to be very similar (sand: 1.01 ± 0.02 m/s versus 1.12 ± 0.02 m/s; grass: 1.1 ± 0.01 m/s versus 1.2 ± 0.01 m/s).

11- L466-471 - the same suggestion as previously described (comment #10) for feeding, drinking, and standing behaviors.

Author Response

Dear reviewer,

Thank you very much for your comments and suggestions. Next, we answer your requirements point by point:

Comments and Suggestions for Authors

The manuscripts by Medina-González et al. describe the relationship between four different flooring types and the movement capacities in dairy cows. The authors used an Analysis of Functional Ranges for posture, kinematics, and kinetics (indicators of movement, behavior, and metabolic capacities). One of the main contributions of the study is the improvement of the welfare and productivity of dairy cows. Furthermore, the knowledge obtained through the review is crucial to cattle producers in general, as lameness is one of the main welfare and health issues faced by the beef industry as well. A strength of the manuscript is the fact of using a large range of indicators to assess the effects of both natural and artificial flooring types on cow's welfare. Overall, the manuscript is very well written. The subject is important and worth reporting, however, in the current form, I find the manuscript a bit superficial and report-like. I encourage the authors to discuss deeper and be critical about the findings of the previous studies. In addition, the authors should describe the potential weaknesses in the study due to the lack of previous studies/references approaching this topic. Below are some suggestions for improvement.

Response: Thanks for the nice description and encouraging comments, we are happy to see that our work was understood. On the suggestion for a deeper and critical discussion of previous findings used as a base for the present manuscript, we believe that the compilation of the data supporting our analysis is enough for the fairly conceptual work we present here (we added up supplementary material to facilitate its revision). Indeed a critical review of every work is needed in order to test our findings, but we feel that to do this, we will need further studies, including controlled experiments, to contrast the information and set up a strong argument to decide if previous works did or did not provided a clear contribution to the cause/effect relationship for movement parameters versus surface in free housing systems. What we present here is definitively a start point for a myriad of experiments that will allow IAFuR modeling tuning. The overarching aim is to look at the bigger picture on health and welfare not only on cattle, but in any type of animal, with both, ecological and evolutionary views. Thats the reason we do not enter in the detailed discussion right now.

1- L139 - please add a reference after: " It allows for a virtually..."

Response: The suggestion is accepted. The paragraph is modified: “This surface prevents sliding, while allowing the homogeneous distribution of the pedal loads at the cow’s feet [8, 14]”. In addition, a reference is included.

2- L142 - please add a reference to the following sentence: "Surface maintains it’s dry..."

Response: The suggestion is accepted. Reference is included.

3- L143 - please add a reference to the following sentence:" A higher walking effort is overcome by the animal."

Response: The suggestion is accepted. Reference is included.

4- L144 - please add a reference to the following sentence: "Its unfavorable compliance under..."

Response: The suggestion is accepted. Reference is included.

5- L147 - please add a reference to the following sentence: "Under wet conditions the asphalt..."

Response: The suggestion is accepted. Reference is included.

6- L152 - please add a reference to the following sentence: "Animal well-being is under the worst conditions."

Response: The suggestion is accepted. Reference is included.

7- L162 - check for double space in the following sentence and throughout the entire manuscript: "____With this information..."

Response: The suggestion is accepted.

8- L181 - " ii) why the grass is the best surface to..." Is this the hypothesis of the study? If yes, please make sure this is clear to the reader. Also, I would like to suggest to the authors to explore more in the introduction section why you believe the grass is the best one by using previous studies as references. 

Response: The suggestion is accepted. The paragraph is modified: “The main questions developed in the present work are: i) what is the movement range for the main flooring types used? and ii) why the grass is the best surface to prevent lameness? We hypothesized that the possibilities of movement are particular for each type of soil used, while grass has an optimal functional range for the cattle in free housing systems.”

9- How concerned are the authors about having such low numbers (n = 13) of studies/references that followed their criteria of selection?

Response: The suggestion is accepted. We have detailed the study selection criteria. The paragraph is modified: “As selection inclusion criteria, we considered that the title and/or abstract of each paper mention movement indicators and at least one of surfaces of interest (i.e. grass, sand, rubber, asphalt or concrete) in dairy cow free-housing systems. Also, the publications selected must include information on units of measurement of at least one of the indicators defined as movement parameters (i.e. posture, kinematics, kinetics, or behavior parameters; see details in Figure 1)”.

In relation to the final number of references used, which might appear low, we can say that the intention of the work is to carry out a systematized review with demanding selection criteria (high detail in delivery of movement parameters according to some type of surface), to reflect the movement of animals according to substrate on the context of dairy cows in free housing and to be able to direct comparison between each other. Finally, as stated in the beginning of this response letter, our present aim is fairly theoretical. The IAFuR approach can only be reinforced and adapted by using further data. The difference now is that, as a result of this research, we do know now what parameters are needed, how to take them and how to interrelated them for a wider health and welfare scope.

10- L444-447 - the authors should add a discussion about the fact that although the speeds described in the previous studies/references were statistically different (if that is the case), biologically they seem to be very similar (sand: 1.01 ± 0.02 m/s versus 1.12 ± 0.02 m/s; grass: 1.1 ± 0.01 m/s versus 1.2 ± 0.01 m/s).

Response: The suggestion is accepted. The paragraph was modified: “Although these differences are statistically different, in all cases the speed exceed the documented biological threshold (> 0.97 m/s) [34], in addition, the stride length presents an optimal functional range on all surfaces (see Supplementary Material 2)”.

11- L466-471 - the same suggestion as previously described (comment #10) for feeding, drinking, and standing behaviors.

Response: The suggestion is accepted. The paragraph is supplemented: “From the biological point of view, these results guarantee a longer rumination time compared to hard and abrasive surfaces (i.e. asphalt and concrete) [53]. In addition, a functional range for welfare and production indicators is shared (see Supplementary Material 2 and Figure 4)”.

Reviewer 2 Report

This manuscript reviewed the effects of different flooring types on various aspects of dairy cow locomotion. It is a timely subject, and the paper includes lots of valuable information and interesting comparisons.

However, for me, concepts like “OFR” lacked clarity and scientific rigor. Perhaps this reflects my ignorance, but the manuscript left me unclear why the authors’ “IAFuR” approach is useful or even really what it is. How was “OFR” calculated? What are the negative impacts of falling outside it? How were numerous additional variables (e.g. animal size, breed, study, etc.) accounted for? When were these results given in the Results and Discussion? In my view, the authors should either explain this approach in much greater detail, or rewrite the manuscript as a more conventional review, evaluating and synthesising results from various studies (which it already does rather well).

I would also like the manuscript to acknowledge some of the complexities surrounding these issues. None of the flooring types discussed are homogenous for all animals across all farms. Pasture quality and management factors, for example, are not mentioned. “Longer time of lying posture” is given as an indicator that a surface is superior, but longer lying durations can indicate poor surfaces (e.g. due to lameness). Recognising such nuances would greatly strengthen the manuscript.

SPECIFIC POINTS

Summary/Abstract:

  • L20: “…of 1 billion DAIRY COWS, and…”
  • L30: “impressive” is the wrong word.
  • L60: This graphical abstract could be clearer and less wordy.

Introduction:

  • L75-76: This point is unclear. Ref?
  • L80: Ref?
  • L96-97: A nit-picky point, but I’m not sure something can be both a “manifestation” and a cause (i.e., “results in”).
  • L110: Ref? Also, both methods arguably can provide early detection (although, as the authors point out, this rarely happens in practice).
  • L113: Since there are various other possible causes, perhaps the authors mean “three MAIN causes”? In which case, ref?
  • L115: This is very specific to stall housing. What about e.g. slipping in the parlour?
  • L121-122: Main factor in what setting? Country? Housing system? Etc.
  • L138, 142, 144, 147, 150, 157: Where?
  • L141, 143, 146, 149, 152: Ref?
  • L141: What about the quality of pasture?
  • L154: This is an example of where structure could be improved. The manuscript moves from discussing lameness prevalence and impacts, to flooring types, and now back to lameness prevalence and impacts. Restructuring to ensure a clear flow of ideas would really strengthen the intro.
  • L155: “too sick” is vague. The animal will likely be euthanised if her input costs exceed her economic benefits.
  • L156: “…the SURFACES with…” (also at L373).
  • L161-162: Please clarify.
  • L172-174: This point could be expanded on.

Methods:

  • L190: When?
  • L198: What were the “initial selection criteria” (if this refers to the previous sentence, please clarify)?
  • L202-204: “cows from Wisconsin”?
  • L310: What research? Refs?

Results and Discussion:

  • L389: Although, again, this effect could well be reversed if poorly-maintained pasture is compared to well-managed indoor systems.
  • L401: Is this based on any peer-reviewed research? The supplied reference, to a veterinary website, is not working.
  • L404: How does this relate to “Navigation capacity”?
  • L441: How does this correspond to the five studies mentioned at L203.
  • L455-456: Why?
  • L474-479: How does this relate to rubber flooring?
  • L482-484: In what sense is “significantly” being used here?
  • L488: Not if it has been raised on rubber flooring?
  • L490-492: Vague.
  • L491-492: Evidence?
  • L515: Can this be inferred if it has not been tested?
  • L540: Or are they manifestations of “negatively affect[ed] health and behaviour”?
  • L650-666: Is this relevant?

Author Response

Reviewer 2

Dear reviewer,

Thank you very much for your comments and suggestions. Next, we answer your requirements point by point:

Comments and Suggestions for Authors

This manuscript reviewed the effects of different flooring types on various aspects of dairy cow locomotion. It is a timely subject, and the paper includes lots of valuable information and interesting comparisons.

However, for me, concepts like “OFR” lacked clarity and scientific rigor. Perhaps this reflects my ignorance, but the manuscript left me unclear why the authors’ “IAFuR” approach is useful or even really what it is. How was “OFR” calculated? What are the negative impacts of falling outside it? How were numerous additional variables (e.g. animal size, breed, study, etc.) accounted for? When were these results given in the Results and Discussion? In my view, the authors should either explain this approach in much greater detail, or rewrite the manuscript as a more conventional review, evaluating and synthesising results from various studies (which it already does rather well).

Response: Thanks for the detailed comments and suggestions. For the development of this theoretical research, we built an Integrative Analysis of Functional Ranges (IAFuR) inspired in a refurbished ME, which allows i) to summarize and systematize quantitative information in qualitative concepts to facilitate the interpretation of movement data; ii) to establish a comprehensive look at the movement phenomena as traditional ME suggests: Movement capacity (how to move?, i.e. posture, kinematics and kinetics) and Navigation capacity (where to move?, i.e. behavior). In addition to traditional MEP, we incorporated to the model the Recovery capacity (when to move? i.e. metabolic cost). Finally, we recommend iii) A decision-making strategy to modify the design and the properties of the soil type (i.e. external factor) according to the functional ranges found.

Using IAFuR, we answered in detail the many aspects that make the grass surface the healthiest one for cattle. But when grass is not available, IAFuR can also help to find the best characteristics for any other optional surface. However, this conceptual approach is in an early stage of development, so the demonstration of such utility requires the inclusion of quantitative information to the model to integrate and weigh the different lines of evidence in order to explain and predict in detail the functional range of movement of animals according to characteristics of the environment (in this case dairy cows).

Response: The suggestion is accepted. The following paragraph is incorporated in methodology, section 2.2: “Functional ranges (either optimal or non-optimal) are a qualitative representation of the relationship between movement, navigation and recovery capacities, in which we superimpose the minimum and maximum values for the movement pathway during the life cycle, highlighting the functional boundaries given by the best-case situation. If the functional range is completely within these limits (i.e. functional boundaries), we call it the optimal functional range (i.e. possibilities of movement on the hypothetical best documented surface; see Graphical Abstract). A range partially outside the boundaries is a sub-optimal functional range. When the range of capacities is completely outside the functional boundaries, it is a non-optimal functional range. Although it is known that the impact of being outside the optimal range is verified in the higher prevalence of lameness (see Supplementary Material 1), the purpose of the IAFuR is to determine the movement profile in each case, as well as to identify the main consequences found for alternative surfaces to grass.  

Figure 3 represents a theoretical outline about the welfare, understood as movement, navigation and recovery adaptive responses, according to usual, fragmented, intervened and disturbed environmental conditions, established as habitats for the study of the animal dispersal [49]. It should be noted that this proposal is in an early stage of development, so for this manuscript specific lines of evidence were used (i.e. movement parameters according to surface) to weigh and integrate results (see Supplementary Material 2), as well as to argue the properties of an alternative surface to grass, if required (see section 3.2). This situation does not restrict the possibility of considering in future quantitative data proposals, additional factors such as animal size, breed, climate and management conditions, among other documented [13, 21-23].

I would also like the manuscript to acknowledge some of the complexities surrounding these issues. None of the flooring types discussed are homogenous for all animals across all farms. Pasture quality and management factors, for example, are not mentioned. “Longer time of lying posture” is given as an indicator that a surface is superior, but longer lying durations can indicate poor surfaces (e.g. due to lameness). Recognising such nuances would greatly strengthen the manuscript.

Response: The suggestion is accepted. A reflection on pasture quality was incorporated in the results and discussion section (response L389): “An interesting question that arises from these results is, do these indicators remain in poorly-maintained pasture compared to well-managed indoor systems? Although we do not have direct evidence about the movement parameters according to different types and quality of grass, if we hypothetically analyze the mechanical properties of the surface, poorly-maintained pasture would lose the optimal balance of the hoof horn wear and growth, in addition to the natural claw load [14]. Therefore, grass surface property in this case, could resemble asphalt (i.e. higher abrasion) affecting the claw conformation [12], as well as concrete (i.e. lower load distribution), causing slipping and “stiff” gait [9, 10]”.

In addition, a reflection on the biological impact of the speed results is incorporated (response L444-7): “Although these differences are statistically different, in all cases the speed exceed the documented biological threshold (> 0.97 m/s) [34], in addition, the stride length presents an optimal functional range on all surfaces (see Supplementary Material 2)”.

Finally we incorporate an analysis for locomotor behavior for the rubber surface (response L466-71): “From the biological point of view, these results guarantee a longer rumination time compared to hard and abrasive surfaces (i.e. asphalt and concrete) [53]. In addition, a functional range for welfare and production indicators is shared (see Supplementary Material 2 and Figure 4)”.

SPECIFIC POINTS

Summary/Abstract:

  • L20: “…of 1 billion DAIRY COWS, and…”

Response: Suggestion is added.

  • L30: “impressive” is the wrong word.

Response: Suggestion is added. It is changed by the concept "critical".

  • L60: This graphical abstract could be clearer and less wordy.

Response: The suggestion is accepted. Removed from figure legend: "This is more common when artificial surfaces are present (rubber, asphalt, concrete), rather than the natural ones (grass and sand), because excretions and cleaning products are easier to be build up”.

The paragraph is modified:” This risk is also notably amplified (light blue) when soaked artificial surfaces (i.e. rubber, asphalt and concrete) become slippery. In the case of soak artificial surfaces (i.e. rubber, asphalt and concrete), it becomes slippery, hence favoring hoof injuries.

Introduction:

  • L75-76: This point is unclear. Ref?

Response: The suggestion is accepted. The phrase is changed to: "Beef and dairy cow production constitute important economic activities worldwide", we also incorporate a reference”.

  • L80: Ref?

Response: The suggestion is accepted. We incorporate a reference.

  • L96-97: A nit-picky point, but I’m not sure something can be both a “manifestation” and a cause (i.e., “results in”).

Response: The suggestion is accepted. The concept "manifestation" is changed to "condition". For the following sentence, it is incorporated: "bovine health problem".

  • L110: Ref? Also, both methods arguably canprovide early detection (although, as the authors point out, this rarely happens in practice).

Response: The suggestion is accepted. We incorporate a reference about the low use of lameness measurement systems in practice. The paragraph is modified: “However, although both methods might provide an early diagnosis, these indicators are rarely been used in large scale farming [20],”

  • L113: Since there are various other possible causes, perhaps the authors mean “three MAIN causes”? In which case, ref?

Response: The suggestion is accepted. The title of the section is changed to “There are three main causes of bovine lameness in free housing systems [9, 23, 26]”. In addition, references 9, 23 and 26 are advanced.

  • L115: This is very specific to stall housing. What about e.g. slipping in the parlour?

Response: The suggestion is accepted. We modified the title of this section to focus all the following ideas to the problems related to lameness associate to free housing system in dairy cows and no others. Through this specification in the title, we resolve the doubts associated with the context (i.e. place) consulted in lines 115, 121-122, 138, 142, 144, 147, 150, and 157).

  • L121-122: Main factor in what setting? Country? Housing system? Etc.

Response: The suggestion is accepted. It is resolved in the response of line 115.

  • L138, 142, 144, 147, 150, 157: Where?

Response: The suggestion is accepted. It is resolved in the response of line 115.

  • L141, 143, 146, 149, 152: Ref?

Response: The suggestion is accepted. It is resolved in the response of line 115.

  • L141: What about the quality of pasture?

Response: The suggestion is accepted. Since no direct evidence is available on this point, it will be addressed preliminary in the results and discussion section (see response L389).

  • L154: This is an example of where structure could be improved. The manuscript moves from discussing lameness prevalence and impacts, to flooring types, and now back to lameness prevalence and impacts. Restructuring to ensure a clear flow of ideas would really strengthen the intro.

Response: The suggestion is accepted. The paragraph is modified:”Asphalt and concrete are the surfaces with the highest levels of lameness prevalence reported. These are the two most commonly used flooring type surfaces due to its low maintenance cost and durability, even when this means having higher losses of their valuable assets [28, 38]. Note that, locomotor disorders account 40% of the unassisted death and euthanasia in dairy cows, because they end up being severely health compromised to keep up with production [37]. Is there a way to reduce bovine losses caused by lameness disease, while keeping maintenance cost low?”

  • L155: “too sick” is vague. The animal will likely be euthanised if her input costs exceed her economic benefits.

Response: The suggestion is accepted. It is changed by the concept: “…severaly health compromised…”.

  • L156: “…the SURFACES with…” (also at L373).

Response: The suggestion is accepted.

  • L161-162: Please clarify.

Response: The suggestion is accepted. The paragraph is modified: “In order to contribute to answer this question we will review the biomechanical characteristics of cow movement and posture, detecting the parameters that fall out of the functional boundaries in all of the five different surfaces described here [30, 39]. Hence, it would be possible to re-adequate management characteristics, including better designed soil surfaces, in order to prevent losses, to improve welfare and production”.

  • L172-174: This point could be expanded on.

Response: It is described in detail in the methodology section, in addition to the application in figure 2. We have also supplemented the paragraph: “This conceptual framework unifies descriptive and predictive models to determine the ecological (environmental) and evolutionary consequences of movement by addressing the questions: why, how, when and where to move? [42]. This inspirational approach allow us to: i) explore the effects of the substrate characteristics on the functional boundaries for movement and posture and ii) suggest a conceptual model raised from the resultant movement parameters (i.e. integrative analysis of functional ranges; IAFuR), to define the optimal properties that an environment should have (e.g. artificial surface design)”.

Methods:

  • L190: When?

Response: The suggestion is accepted. The paragraph is modified: “A literature review was carried out during the months of September, 2017 and December, 2018; taking into account kinematics, kinetics, behavior and posture parameters according to the main surfaces used for free housing dairy cows”.

  • L198: What were the “initial selection criteria” (if this refers to the previous sentence, please clarify)?

Response: The suggestion is accepted. The paragraph is modified: “As selection inclusion criteria, we considered that the title and/or abstract of each paper mention movement indicators and at least one of surfaces of interest (i.e. grass, sand, rubber, asphalt or concrete) in dairy cow free-housing systems. Also, the publications selected must include information on units of measurement of at least one of the indicators defined as movement parameters (i.e. posture, kinematics, kinetics, or behavior parameters; see details in Figure 1)”.

  • L202-204: “cows from Wisconsin”?

Response: The suggestion is accepted. In the manuscripts of Cook et al. (references 27 and 61) Wisconsin cow breeds are not specified. The following is incorporated: “breed not specified”.

  • L310: What research? Refs?

Response: The suggestion is accepted. Reference is included.

Results and Discussion:

  • L389: Although, again, this effect could well be reversed if poorly-maintained pasture is compared to well-managed indoor systems.

Response: The suggestion is accepted. The paragraph is modified: “An interesting question that arises from these results is, do these indicators remain in poorly-maintained pasture compared to well-managed indoor systems? Although we do not have direct evidence about the movement parameters according to different types and quality of grass, if we hypothetically analyze the mechanical properties of the surface, poorly-maintained pasture would lose the optimal balance of the hoof horn wear and growth, in addition to the natural claw load [14]. Therefore, grass surface property in this case, could resemble asphalt (i.e. higher abrasion) affecting the claw conformation [12], as well as concrete (i.e. lower load distribution), causing slipping and “stiff” gait [9, 10]”.

  • L401: Is this based on any peer-reviewed research? The supplied reference, to a veterinary website, is not working.

Response: The suggestion is accepted. Reference is changed: Telezhenko, E. Effect of flooring system on locomotion comfort in dairy cows. Diss. (sammanfattning/summary) Skara : Sveriges lantbruksuniv., Acta Universitatis Agriculturae Sueciae, 1652-6880; 2007:76. ISBN 978-91-576-7375-6. [Doctoral thesis] https://pub.epsilon.slu.se/1558/

  • L404: How does this relate to “Navigation capacity”?

Response: The suggestion is accepted. The idea is reinforced in the paragraph: “In sand surface, only the stand up/lie down locomotory behavior of dairy cows is considered as an indicator of navigation capacity, because it answers the questions: when and where to move? [42]. Contrary to the normal condition found in ME concept [42], the animals certainly are not really free to move wherever they want, indeed, they are kept in a closed environment. Therefore, most spatial distribution is lead and controlled by the farmer, but even within this restriction, the animal still can decide to stand-up or lie down”.

  • L441: How does this correspond to the five studies mentioned at L203.

Response: The suggestion is accepted. The number of studies in the L203 is corrected: “…rubber was analyzed in 10 investigations…”

  • L455-456: Why?

Response: The suggestion is accepted. The paragraph is modified: “Surely, this unnatural feeling is produced by a proprioceptive reaction response to substrate competence (i.e. different mechanical properties) [67]”.

  • L474-479: How does this relate to rubber flooring?

Response: The suggestion is accepted. Reference is made to Supplementary Material 2, Additional Figure 4.

  • L482-484: In what sense is “significantly” being used here?

Response: The suggestion is accepted. The word “significantly” is omitted.

  • L488: Not if it has been raised on rubber flooring?

Response: The suggestion is accepted. The paragraph is modified: “This artificial surface makes bovines less confident [76]”.

  • L490-492: Vague.

Response: The suggestion is accepted. To complement information, refer to section 3.2.1. Surface material properties.

  • L491-492: Evidence?

Response: The suggestion is accepted. To complement information, refer to section 3.2.1. Surface material properties.

  • L515: Can this be inferred if it has not been tested?

Response: The suggestion is accepted. The concept “infer” is changed to “hypothesize”.

  • L540: Or are they manifestations of “negatively affect[ed] health and behaviour”?

Response: The suggestion is accepted. The paragraph is modified: “These indicators favor the appearance of foot lesions, hemorrhages, dermatitis, sole erosions, which present lameness as a functional consequence [74, 75].”

  • L650-666: Is this relevant?

Response: The suggestion is accepted. We deleted the paragraph.

Round 2

Reviewer 2 Report

I thank the authors for comprehensively addressing my earlier concerns. The manuscript is now excellent, and I only have a few minor edits.

*L102: "determinates" > "determines". Also, "IT is considered...".

*L116-119: Grammar.

*L507: "exceedS"